# Amdahl's Law for LLMs: A Throughput-Centric Analysis of Extreme LLM Quantization

**Jinendra Malekar**                                        *jmalekar@email.sc.edu*
*Department of Computer Science and Engineering*
*University of South Carolina*

**Ramtin Zand**                                        *ramtin@cse.sc.edu*
*Department of Computer Science and Engineering*
*University of South Carolina*

**Reviewed on OpenReview:** *https: // openreview. net/ forum? id= JtrQJJQYpP*

## Abstract

The emergence of 1-bit large language models (LLMs) has sparked significant interest, promising substantial efficiency gains through extreme quantization. However, these benefits are inherently limited by the portion of the model that can be quantized. Specifically, 1-bit quantization typically targets only the projection layers, while the attention mechanisms remain in higher precision, potentially creating significant throughput bottlenecks. To address this, we present an adaptation of Amdahl's Law specifically tailored to the LLMs, offering a quantitative framework for understanding the throughput limits of extreme quantization. Our analysis reveals how improvements in quantization can deliver substantial throughput gains, but only to the extent that they address critical throughput-constrained sections of the model. Through extensive experiments across diverse model architectures and hardware platforms, we highlight key trade-offs and performance ceilings, providing a roadmap for future research aimed at maximizing LLM throughput through more holistic quantization strategies.

## 1 Introduction

Generative Large language models (LLMs) such as GPT (Radford et al., 2019), OPT (Zhang et al., 2022) and LLaMA (Touvron et al., 2023) have attracted significant attention in recent years because of their impressive performance across various tasks, including but not limited to machine translation (Xu et al., 2024a), conversational chatbots (Sánchez Cuadrado et al., 2024), question answering (Tan et al., 2023), and even code generation (Ugare et al., 2024).

However, the remarkable performance of LLMs has come with increasing computational and energy costs. This necessitates expanding high-performance computing resources in data centers, potentially delaying green energy commitments and causing adverse environmental impacts (Stojkovic et al., 2024). Consequently, recent research has focused on optimizing the energy footprint of LLMs through various model optimization and compression techniques like pruning (Ma et al., 2023) and quantization (Xiao et al., 2023; Shao et al., 2023; Ma et al., 2024).

In addition to enhancing the energy efficiency of running LLMs in data centers, model optimization and compression can facilitate the deployment of LLMs on mobile and edge computing devices for real-time processing (Reidy et al., 2023; Ardakani et al., 2025). This advancement could benefit various emerging applications, including social robotics (Addlesee et al., 2024; Esteban-Lozano et al., 2024), and augmented and virtual reality (Asadi et al., 2024; Morales & Showalter-Bucher, 2023).

Among LLM compression approaches, quantization has become a focal point, driven by works such as Smoothquant (Xiao et al., 2023), Omniquant (Shao et al., 2023), and more recently ShiftAddLLM (You

et al., 2024) which emphasize post-training quantization (PTQ) as a cost-effective approach avoiding the processes of retraining and fine-tuning which can be particularly costly for LLMs.

A newly emerging approach to optimizing and compressing models through quantization-aware training (QAT) involves the extreme quantization of certain portions of LLMs, using binary $\{-1, 1\}$ and ternary $\{-1, 0, 1\}$ weights (Wang et al., 2023; Ma et al., 2024). This development has initiated the era of 1-bit LLMs. Besides the memory utilization advantages, extreme quantization transforms the costly matrix multiplication (MatMul) operations into more efficient addition and subtraction operations, and thus leading to MatMul-free operations. However, it is important to note that not all MatMul operations can undergo extreme quantization due to the resulting drop in accuracy. Specifically, MatMul operations in the attention heads still require higher precisions, such as 16-bit floating point (FP16) or 8-bit integer (INT8). A follow-up work (Zhu et al., 2024) has built upon BitNet (Wang et al., 2023), aiming to eliminate MatMul operations in attention heads by using Hadamard products.

The advent of 1-bit LLMs has paved the way for various new research directions. However, since they currently address only a fraction of the model, a crucial question arises:

> *What effects do partial enhancements in 1-bit LLMs have on the overall performance of the model?*

Answering this question is vital for guiding future research effectively and avoiding misguided or ineffective goals. For example, if the MatMul operations that are replaced with MatMul-free operations in current 1-bit LLMs account for the majority of the model's computation and memory usage, then focusing on optimizing the relatively minor MatMul operations in attention heads may be less impactful. In this case, prioritizing custom hardware development to fully leverage extreme quantization would be more sensible. Conversely, if the conversion of the fraction of MatMul operations to MatMul-free operations in current 1-bit LLMs does not significantly affect overall computation and memory usage, it would be more prudent to focus on optimizing the MatMul operations in the attention heads—currently not optimized in 1-bit LLMs—rather than investing in hardware development for current 1-bit LLMs.

In this study, we address the highlighted research question through extensive experiments and analyses on various LLMs. We explore various model hyperparameters across two types of hardware designed for edge and cloud environments. In addition, we propose an adaptation of Amdahl's Law for LLMs to identify how partial improvements in LLM can translate into overall enhancements for the entire model. Our findings reveal important nuances that can guide future research in the 1-bit LLMs era.

## 2 Related Work

### 2.1 Transformer Quantization

Transformer quantization can be categorized into weight-only and weight-and-activation quantization. Weight-only quantization reduces memory requirements, while weight-and-activation quantization also enhances computational efficiency. SmoothQuant (Xiao et al., 2023) supports both activation and weight quantization, based on the principle that the difficulty of activation quantization can be mathematically transformed. This method demonstrates a 1.5× speedup and 2× memory reduction for LLMs with negligible loss, supporting W8A8 (8-bit weight, 8-bit activation) quantization. Omniquant (Shao et al., 2023) is another method supporting a broader spectrum of weight-activation quantization configurations, including W4A4, W4A16, W3A16, and W2A16. This is achieved through learnable weight clipping, which optimizes the clipping threshold, and learnable equivalent transformation, which mitigates activation outliers by shifting them to weights. LLM.int8() (Dettmers et al., 2022) loads the model in 8-bit format, with 99.9% of operations performed in 8-bit, except for emergent outliers. It uses vector-wise quantization with different normalization constants to preserve model performance. GPTQ (Frantar et al., 2023) focuses solely on weight quantization, achieving 3-bit and 4-bit quantization with speedups of 3.24× and 4.53× on A6000 and A100 GPUs. QuIP# (Tseng et al., 2024) is a weight-only quantization method that achieves state-of-the-art performance in sub-4-bit quantization. It employs a randomized Hadamard transform combined with vector quantization and then fine-tunes the model to enhance its fidelity. All the methods discussed above are related to PTQ.

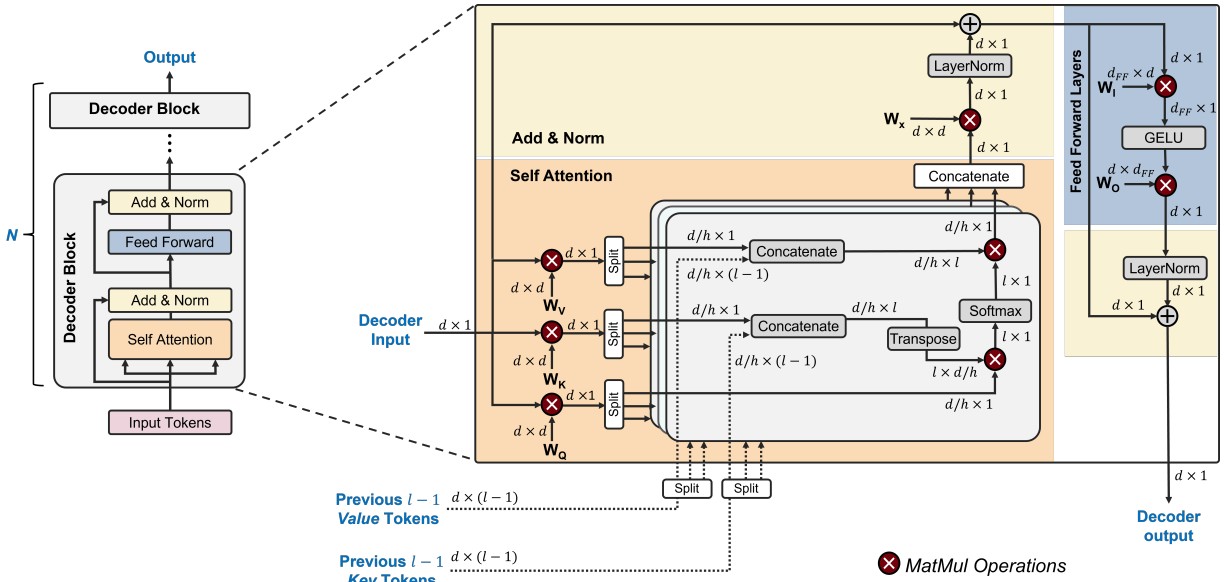

Figure 1: The typical architecture of decoder-only LLM models and its underlying operations. The tokenization and embedding layers are not shown in the figure.

## 2.2 Era of 1-bit LLMs

Before the LLM era, the concept of binary weight and activation quantization was explored in works such as Binarized Neural Networks (BNN) (Hubara et al., 2016) and Ternary Neural Networks (TNN) (Alemdar et al., 2017). While these studies did not focus on generative language tasks, they achieved significant performance improvements, with BNNs demonstrating up to $7\times$ performance gains and TNNs showing up to $3.1\times$ energy efficiency improvements. These findings underscore the remarkable ability of neural networks to operate effectively with just 1-bit precision.

Recently, following the rise of LLMs and advancements in model performance, the BitNet (Wang et al., 2023) introduced a 1-bit transformer quantization technique for LLMs. This method replaces the conventional 'nn.Linear' layer in PyTorch with a new BitLiner layer, where weights are restricted to either 1 or -1, and activations are represented in 8-bit precision. Despite this quantization, other components like self-attention remain in 8-bit format. BitNet's design suggests that it can scale to even larger transformer models, following scaling laws similar to those used for full-precision transformers. A variant of BitNet, known as BitNet 1.58 (Ma et al., 2024), employs ternary weights (-1, 0, 1) and achieves perplexity and end-task performance comparable to full-precision transformers (FP16 or BF16). In terms of the computation, 1-bit LLMs transform MatMul operations into addition operations, due to the 1-bit nature of the weights, except for layers like attention, which need to remain in high precision to maintain performance. Additionally, 1-bit LLMs address the challenge of transferring model parameters from DRAM to on-chip accelerators such as SRAM, prompting the development of architectures optimized for the efficient operation of 1-bit LLMs.

## 3 Approach

### 3.1 Demystifying the Underlying Operations of LLMs

The overall architecture of the generative LLMs is shown in Figure 1, which includes $N$ decoder blocks, each consisting of self-attention and feedforward layers followed by add and normalization operations (Vaswani et al., 2017). The core of the LLM is the self-attention mechanism with multiple heads ($h$). For an attention head, the computation begins with three linear projections of the token vector to form the Key ($K$), Query ($Q$), and Value ($V$) vector sequences:

$$Q = W_Q * I, \quad K = W_K * I, \quad V = W_V * I \tag{1}$$

| Block | Description | Dimension |
|---|---|---|
| Attention Projections | $W_Q$ 
 $W_K$ 
 $W_V$ 
 $W_X$ | $(d \times d) * (d \times 1)$ |
| Attention Head | $Q * K$ 
 $V * Score$ | $(l \times d/h) * (d/h \times 1)$ 
 $(d/h \times l) * (l \times 1)$ |
| FFN | Intermediate FF 
 Output FF | $(d_{FF} \times d) * (d \times 1)$ 
 $(d \times d_{FF}) * (d_{FF} \times 1)$ |

Table 1: MatMul Operations and their dimensions in decoder-only LLMs. The parameters $d$, $h$, $l$, and $d_{FF}$ denote embedding dimension, number of attention heads, sequence length, and the size of the feed-forward layers, respectively.

where $I$ is the input token vector and $W_Q$, $W_K$, and $W_V$ are trainable weight matrices with $d \times d$ dimensions where $d$ is the embedding dimension. The operator $*$ denotes the MatMul operation.

The generated $K$, $Q$, and $V$ vectors are then divided into $h$ vectors with reduced dimensionality of $d/h$ where $h$ is the number of attention heads. At this stage, the value and key vectors are concatenated with the previous $l-1$ value and key tokens that are cached from previous token generation iterations to form a $d/h \times l$ matrix in each head, where $l$ is the sequence length. Next, the attention scores are computed using the scaled dot-product of the queries and keys multiplied with the generated value matrix (Vaswani et al., 2017). This step includes two MatMul ($*$) operations:

$$\text{Attention}(Q, K, V) = softmax(\frac{Q * K^T}{\sqrt{d}}) * V \tag{2}$$

Subsequently, the output of the attention heads are concatenated and linearly transformed as follows:

$$\text{MultiHead}(Q, K, V) = \text{Concat}(head_1, ..., head_h) * W_X \tag{3}$$

where $head_i = \text{Attention}(Q_i, K_i, V_i)$, and $W_X$ is a $d \times d$ matrix with trainable elements.

Finally, the attention output is followed by a feed-forward network (FFN) that involves two linear and one nonlinear transformations. As shown in Figure 1, Gaussian error linear unit (GELU) (Hendrycks & Gimpel, 2016) activation function is frequently utilized in LLMs to introduce nonlinearity into FFN. Therefore, the FFN computation can be described as follows, which involves two more MatMul operations:

$$FFN(x) = GELU(x * W_I + b_I) * W_O + b_O \tag{4}$$

where $W_I$ and $W_O$ are trainable matrices with $d_{FF} \times d$ and $d \times d_{FF}$ dimensions, respectively, where $d_{FF}$ is the feed-forward layer size. Also, the $x$ is MultiHead $(Q, K, V)$. Both the self-attention and FFN layers are followed by layer normalization which can also introduce some nonlinearity in the process of division by the standard deviation.

In summary, the computation of decoder blocks in LLMs involves a combination of nonlinear operations (LayerNorm, softmax, GELU), and linear MatMul operations. In particular, there are a total of $2h + 6$ MatMul operations in a decoder block including six weight-to-activation MatMuls ($W_Q$, $W_K$, $W_V$, $W_X$, $W_I$, and $W_O$ projections) plus $2h$ activation-to-activation MatMuls to compute $Attention(Q, K, V)$ in each head ($h$). In decoder-only LLMs, MatMuls are matrix-vector multiplications since the inference is performed iteratively, processing one input token per iteration and the keys and values from previous iterations are cached as shown in Figure 1. Table 1 lists the dimensions of each of these MatMul operations.

Previous works (Kim et al., 2023) have shown that designing dedicated hardware to compute nonlinear operations in LLMs can make their computation overhead negligible compared to MatMul operations. Consequently, optimizing MatMul operations can lead to a significant speedup in LLM computation. The 1-bit

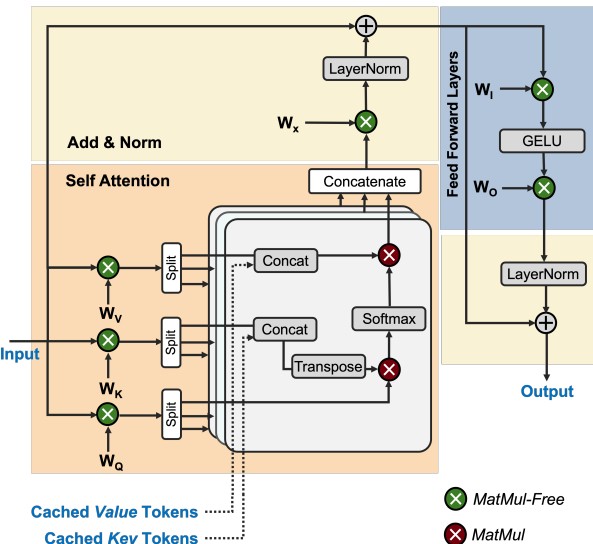

Figure 2: The 1-bit LLMs divide the model into two portions: attention heads with MatMul operations (shown in red) and MatMul-free projection layers (shown in green).

LLMs (Wang et al., 2023; Ma et al., 2024; Xu et al., 2024b) as a recent approach that has attracted considerable attention involves extreme quantization of MatMuls in LLMs, except for the attention heads, which require higher precision to maintain accuracy. Quantizing all the weight matrices ($W_Q$, $W_K$, $W_V$, $W_X$, $W_I$, and $W_O$) in the LLMs to include only binary $\{-1, 1\}$ or ternary $\{-1, 0, 1\}$ elements transforms the weight-to-activation MatMul operations to simple addition and subtraction operations, dividing the LLM models into two portions one with MatMul and one without MatMul (or MatMul-free), as shown in Figure 2.

Due to variations in the model hyperparameters ($d$, $l$, $h$, and $d_{FF}$) across different types of LLMs, the proportion of the linear projections (weight-to-activation MatMuls) and attention head (activation-to-activation MatMuls) computation relative to the entire model can vary significantly. Therefore, performance analysis with layer-wise granularity, targeted in this work, can illuminate the effectiveness of the 1-bit LLMs and help determine future research directions.

## 3.2 Design of LLM-Specific Hardware

While GPUs are commonly used for LLM inference and training due to their general-purpose parallelism, we focus on TPUs to explore design trade-offs specific to LLM workloads. Unlike GPUs, TPUs allow us to directly control memory hierarchy and dataflows critical for modeling 1-bit LLM performance. Results from GPU-based setups, discussed in Appendix D, further contextualize our findings.

For the performance analysis, we leverage TPU architectures which are specifically designed to accelerate the MatMul operations dominant in machine learning workloads. TPUs maximize data reuse while minimizing data transfer by utilizing systolic arrays at their core (Jouppi, 2017; Jouppi et al., 2017).

A systolic array typically consists of two-dimensional arrays of processing elements (PEs). Each PE performs a multiply-and-accumulate (MAC) operation, multiplying weights and inputs using a multiplier circuit and adding the result to previously computed partial sums using an accumulator circuit. The MAC result is either retained within the same PE or broadcast to other PEs for further computations, depending on the dataflow architecture. This MAC operation is executed in every PE of the systolic array, enabling efficient MatMul operations by maximizing data reuse and minimizing additional data transfer overhead.

The dataflow in a systolic array is a mapping scheme determined by the microarchitecture of the PEs, which dictates how input data is fed into the array and how partial results and outputs are generated and stored. Rather than repeatedly loading and storing data to and from memory, each PE typically follows one of

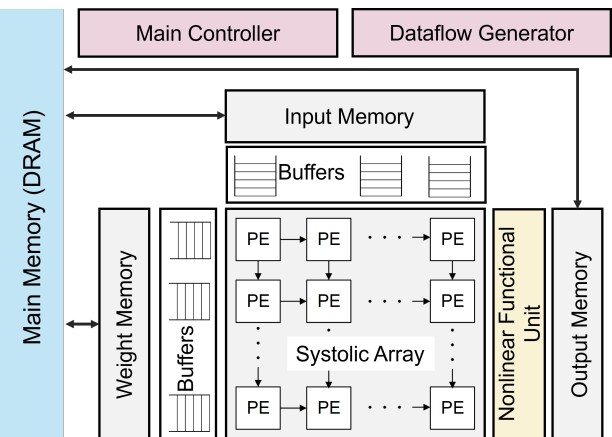

Figure 3: The overall architecture of the TPU that is designed for accelerating LLMs, featuring dedicated hardware to support nonlinear operations.

these dataflow architectures: *(1) Input Stationary (IS)*: The inputs (or activations) remain fixed in the PEs while the weights are sequentially fed into the PEs; *(2) Output Stationary (OS)*: The outputs are attached with the MAC units, while the inputs and weights circulate among the PEs. Inputs and weights are loaded, multiplied, and the results are accumulated with partial sums held in the PE. *(3) Weight Stationary (WS)*: Each weight is preloaded into a register within each PE. During each cycle, inputs are multiplied by the weights and broadcast across the PEs.

Figure 3 shows the architecture of TPU, consisting of weight, input, and output memories, and a systolic array of size $S = N \times N$ PEs surrounded by the first-in-first-out (FIFO) buffers. Additionally, our TPU includes a *Nonlinear Functional Unit*, featuring custom hardware to support nonlinear operations in the LLMs. The *Dataflow Generator* block generates the memory read/write addresses to store or retrieve the inputs, weights, and outputs according to the selected dataflow. The *Main Controller* manages the data transfer between memories, FIFOs, and the systolic array.

As previously discussed, the MatMul operations in generative LLMs involve matrix-vector multiplications. Consequently, the sizes of the input and output vectors are always smaller than those of the weight matrices. For activation-to-activation MatMuls in the attention head, where there are no weight values (See Figure 1), we store the concatenated *Value* and *Key* matrices (with $d/h \times l$ and $l \times d/h$ dimensions, respectively) in the weights memory, while the *Query* and *attention score* vectors are stored in the input memory. Based on the pattern in the size of the input, weight, and output tensors in matrix-vector multiplications involved in LLMs (mentioned in Table 1), a TPU design with larger weight memory compared to input and output memories would be more efficient, as it reduces the need for costly accesses to the main DRAM memory to load the weights.

For the dataflow architecture, we conducted comprehensive experiments utilizing IS, OS, and WS dataflows. Based on the results obtained (refer to Appendix A ), the OS dataflow architecture demonstrated the best performance. The OS dataflow is particularly advantageous for accumulating results since partial sums remain stationary and do not need to be moved frequently. Additionally, once weight and input values are fetched from their respective memories, they are reused by passing from one PE to another, leveraging the spatial dataflow capabilities of TPUs.

### 3.3 Amdahl's Law of LLMs

Since current 1-bit LLMs only improve a part of the model (the projections without enhancing the attention heads), we need a mechanism to determine how these partial improvements translate into overall enhancements for the entire model. This is crucial for addressing the main research question of the paper.

The Amdahl's Law provides a framework for such scenarios. Amdahl's Law is a formula used to find the maximum improvement in a system when only part of it is improved. It can be expressed as:

$$S_{total} = \frac{1}{1 - F + \frac{F}{S_{partial}}} \qquad (5)$$

where $F$ is the fraction of the system that is improved, $S_{partial}$ is the factor by which the part $F$ is improved, and $S_{total}$ is the overall improvement of the entire system.

In the context of 1-bit LLMs, we define $F$ as the fraction of the MatMul operations that can be replaced with MatMul-free operations by using extreme quantization, relative to all MatMul operations in the model. This formulation enables practitioners to reason about speedup potential ($S_{partial}$) as a function of hardware-specific parameters (e.g., systolic array size) and model hyperparameters such as embedding size d, context length l, feedforward dimension dFF, and number of attention heads h. Here, our focus is on quantifying the value of $F$ across various LLMs and hardware configurations. Enhancing $S_{partial}$ is closely tied to the design of custom hardware accelerators for binary and ternary operations, which is beyond the scope of this paper. Nonetheless, to provide context, a recent processing-in-memory (PIM)-based hardware design (Malekar et al., 2025) targeting the acceleration of projection layers of 1-bit LLMs demonstrated up to an $80\times$ increase in throughput. In Appendix F, we provide a more generalized variation of the proposed Amdahl's Law of LLM to enable a more general and hardware-agnostic analysis.

## 4 Experiments

### 4.1 Simulation Setup

For our experiments, we study 13 different LLMs including GPT, OPT, and LLaMA models. Our work assumes **W1A8**, **W2A8** quantization as the baseline quantization schemes throughout the analysis. Specifically, projection weights are represented using binary or ternary quantization (i.e., 1-2 bits for weights), while activations are maintained at 8-bit integer precision (INT8). This reflects the implementation principles of systems like BitNet (Wang et al., 2023) and BitNet 1.58 (Ma et al., 2024), which preserve accuracy by keeping activations in moderate precision while aggressively quantizing weights. Table 2 lists all models and their corresponding hyperparameters. To save space in the main body of the paper, we only provide the results for the seven OPT models, as they represent a diverse range of model hyperparameter combinations. The results for GPT and LLaMA models are provided in the Appendix B.

For the hardware, we designed two TPUs tailored for different applications: cloud and edge processing. The cloud TPU features a $256 \times 256$ systolic array with 16MB of SRAM, while the edge TPU has a $32 \times 32$ systolic array with 8MB of SRAM. Both systolic arrays employ an OS dataflow. Also, in both designs, 2MB of memory is allocated for internal use, including storing control and configuration data, tracking computation states, managing data flow, and ensuring seamless data movement. All memories use double buffering to mask the latency associated with SRAM access. Table 3 provides the memory distribution of both edge and cloud TPU designs. We utilize the cycle-accurate SCALE-Sim framework (Samajdar et al., 2018; 2020) to measure compute cycles and memory accesses in various LLMs.

Our experiments investigate the distribution of computation and memory utilization between the attention heads, which require MatMul operations, and the projections, which can be binarized or ternarized and consequently do not involve MatMuls. In the results presented, the terms "MatMul" and "MatMul-Free" refer to these respective parts of the LLM computation.

### 4.2 Performance Analysis on Cloud Setup

In the first set of experiments, we examine various language models deployed on the cloud TPU setup to determine the fraction of the models that can become MatMul-Free in 1-bit LLMs, i.e., projection layers, through extreme quantization. To achieve this, we compare seven OPT models of various sizes (ranging from 350M to 66B parameters) with different sequence lengths (ranging from 128 to 4096). We vary the

| Models | Param | Hyperparameters | | | |
|---|---|---|---|---|---|
| | | $d$ | $h$ | $d_{FF}$ | $l$ |
| GPT | 125M | 768 | 12 | 768 | 128-4096 |
| | 355M | 1024 | 16 | 1024 | 128-4096 |
| | 774M | 1280 | 20 | 1280 | 128-4096 |
| | 1.5B | 1600 | 25 | 1600 | 128-4096 |
| OPT | 350M | 1024 | 16 | 4096 | 128-4096 |
| | 1.3B | 2048 | 32 | 8192 | 128-4096 |
| | 2.7B | 2560 | 32 | 10240 | 128-4096 |
| | 6.7B | 4096 | 32 | 16384 | 128-4096 |
| | 13B | 5120 | 40 | 20480 | 128-4096 |
| | 30B | 7168 | 56 | 28672 | 128-4096 |
| | 66B | 9216 | 76 | 36864 | 128-4096 |
| LLaMA | 7B | 4096 | 32 | 11008 | 128-4096 |
| | 13B | 5120 | 40 | 13824 | 128-4096 |

Table 2: Model Hyper-parameters used for simulations.

| Design | Systolic Array Size | Memory Capacity | | |
|---|---|---|---|---|
| | | Input | Output | Weight |
| Cloud | $256 \times 256$ | 4MB | 4MB | 8MB |
| Edge | $32 \times 32$ | 2MB | 2MB | 4MB |

Table 3: The specification of the edge and cloud TPUs.

sequence length ($l$) because it only affects the computation in the attention heads and determines the size of the remaining MatMul operations in 1-bit LLMs (refer to Table 1).

Figure 4 exhibits the fraction of MatMul-Free operations in the OPT models deployed on the cloud setup, measured in terms of compute cycles and memory accesses. The fraction of MatMul-Free operations varies from roughly **23%** to **98%** for compute cycles and from **59%** to **99.8%** for memory access across different configurations. **In general, MatMul-Free operations increase with model size and decrease with sequence length.**

### 4.2.1 Compute Cycle Analysis.

Typically, the context length of the OPT models is equal to 2048. The MatMul-Free compute cycles for these OPT models can be observed in the 2048 row of Figure 4 (a). For smaller language models like OPT 350M, approximately 63% of the computation occurs in the attention heads, involving MatMul operations, while only 37.1% of the computation benefits from MatMul-Free operations provided by 1-bit LLMs. Moreover, the OPT 1.3B model with a sequence length of 2048 is particularly noteworthy, as half of its computation can be MatMul-Free, while the other half involves MatMul operations.

Models like LLaMA 7B and 13B generally use larger sequence lengths of 4096. In terms of model size, they are comparable to OPT 6.7B and 13B models. As illustrated in the 4096 row of Figure 4 (a), MatMul-Free operations account for 64.5% and 69% of the computation for OPT 6.7B and OPT 13B models, respectively. A detailed analysis of the LLaMA models can be found in the Appendix B.

### 4.2.2 Memory Access Analysis.

1-bit LLMs, such as BitNet (Wang et al., 2023) and BitNet 1.58 (Ma et al., 2024), can achieve up to $16\times$ and $8\times$ reductions in weight memory utilization compared to FP16 LLMs because they use just 1 bit and 2 bits to represent binary and ternary weights, respectively. Besides memory capacity, another crucial factor affecting LLM throughput is the number of memory accesses required for inference. Therefore, we also analyzed the

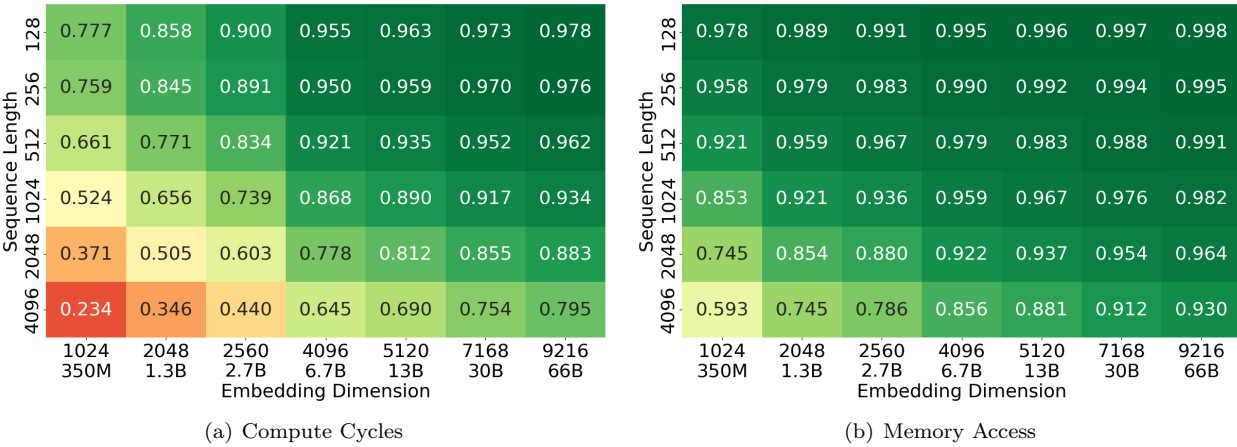

(a) Compute Cycles

(b) Memory Access

Figure 4: Fraction of MatMul-free operations in the OPT models deployed on the cloud setup.

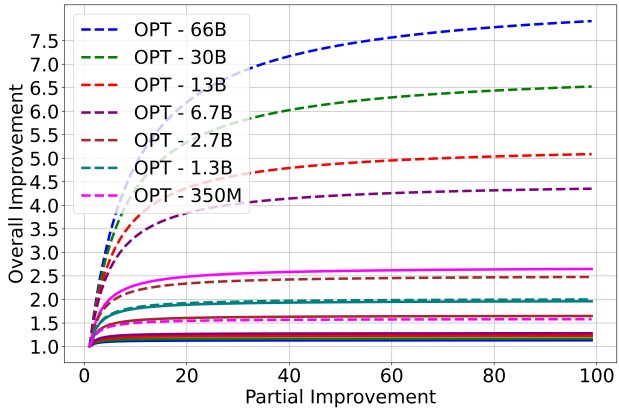

Figure 5: Amdahl's Law of LLMs for cloud deployment scenario. The dashed lines and solid lines show the effect of partial improvement in projection layers and partial improvement in attention layers, respectively.

memory access of the MatMul-Free and MatMul components of the 1-bit LLM architecture. Figure 4 (b) shows the ratio of memory reads and writes associated with the projection layers (MatMul-Free parts) to those of the entire model.

The results reveal a trend similar to the compute cycle analysis. However, unlike compute cycles, for all cases, the majority of memory accesses are associated with the projection layers. For example, for OPT models with a sequence length of 2048, the fraction of memory access for the OPT 350M is **74%**, increasing to **96%** for the OPT 66B. The memory access analysis suggests opportunities for research into hybrid memory hierarchy designs, where the components of the model dominating memory accesses can be offloaded to faster memory technologies. For example, in OPT 6.7B with a sequence length of 4096, moving the projection layers' weight and activation data to faster memory can speed up more than 85% of all memory accesses.

### 4.2.3 Amdahl's Law of LLMs.

Here, we leverage Amdahl's Law of LLMs proposed in Equation (5) to show how partial improvements in the LLMs can enhance overall performance. In particular, we vary $S_{partial}$ from 1 to 100, and using the fractions ($F$) shown in Figure 4, calculate the overall improvement of model ($S_{total}$). Figure 5 demonstrates an Amdahl's Law analysis when improvements are applied to either attention layers (MatMul parts) or the projections layers (MatMul-Free parts) of LLMs. The dashed lines show the effects of improving the projection layers, while the solid lines represent the impact of improvements to the attention layers. This

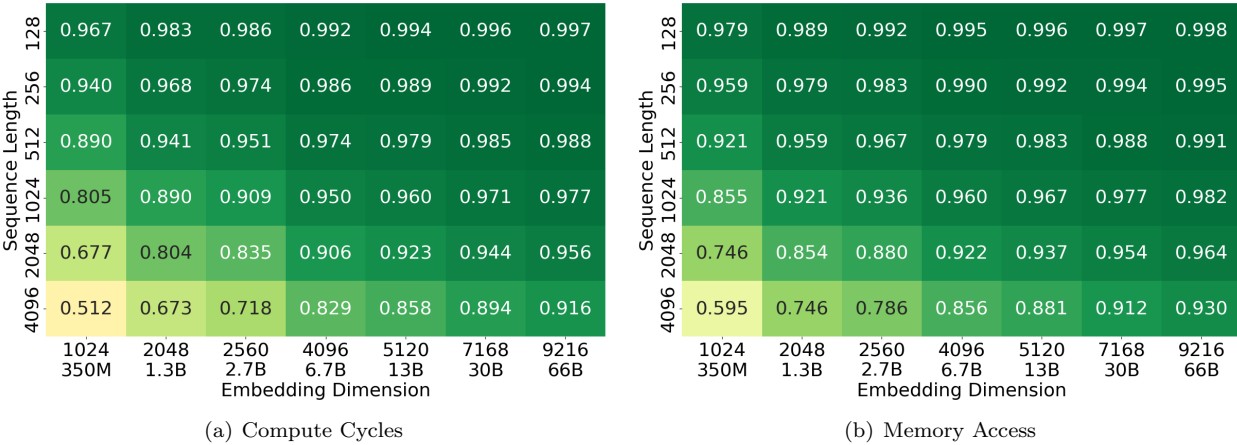

Figure 6: Fraction of MatMul-free operations in the OPT models deployed on the edge setup.

analysis is based on OPT models with a typical sequence length of 2048. For Amdahl's Law analyses with other sequence lengths, please refer to the Appendix C.

The Amdahl's Law analysis of LLMs deployed on the cloud setup reveals three key takeaways:

1. For smaller language models like OPT 350M, improving the attention heads has a greater impact than enhancing the projection layers. Therefore, using 1-bit LLM paradigms to improve these models may result in limited overall performance gains.

2. Medium-sized LLMs, such as OPT 1.3B and 2.7B, can benefit from combining 1-bit LLM improvements with enhancements to the attention heads, such as replacing MatMul operations with Hadamard products and additive operators as proposed in (Zhu et al., 2024).

3. For larger models like OPT 6.7B and above, improvements to the attention heads have a minimal effect on overall performance. In these cases, employing 1-bit LLM methods alone can lead to significant gains in performance and throughput.

## 4.3 Performance Analysis on Edge Setup

Figure 10 presents the compute and memory analysis for the edge setup. In all cases, most of the computations occur in the MatMul-Free portion. The only instance where the computation is relatively evenly distributed between the MatMul and MatMul-Free parts is for OPT 350M with a 4096 sequence length. A similar pattern is observed in memory accesses. This happens because, in the edge setup, the smaller $32 \times 32$ systolic array is less efficient at handling the larger matrices typically found in projection layers. Meanwhile, the MatMul operations in the attention heads are smaller due to the splitting of large matrices among multiple heads (refer to Figure 1), making these operations more manageable even with smaller systolic arrays. Consequently, this increases the ratio of computation in the projection layers compared to the attention heads.

The difference between the cloud and edge setups stems from their scaling characteristics. In cloud TPUs ($256 \times 256$ arrays), large MatMuls in projection layers are efficiently handled due to ample compute and memory bandwidth, leading to pronounced compute savings for MatMul-free sections. In contrast, edge TPUs ($32 \times 32$ arrays) suffer degraded efficiency on large projections due to limited parallelism and SRAM, capping compute benefits despite similar memory access trends.

Additionally, the cloud setup faces underutilization of processing elements when executing smaller MatMuls in attention heads, whereas the edge setup rarely encounters this issue. This helps explain the observed difference in the compute cycle fraction ($F$) between edge and cloud environments, even though the memory

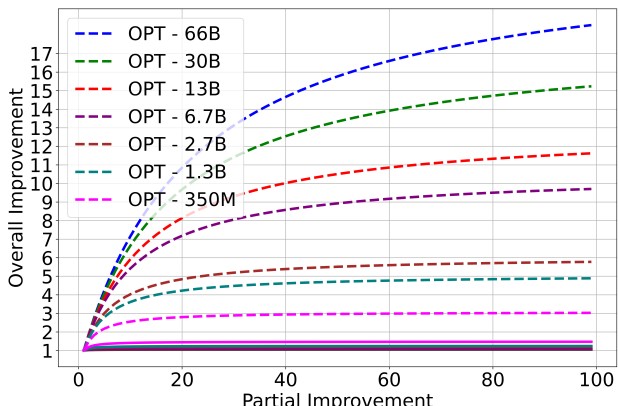

Figure 7: Amdahl's Law of LLMs for edge deployment scenario. The dashed lines and solid lines show the effect of partial improvement in projection layers and partial improvement in attention layers, respectively.

access patterns remain largely similar across both. We include additional experiments in Appendix E, where the embedding dimension is fixed (d = 4096) and the number of attention heads is varied, to isolate the impact of attention head MatMul dimensions on the compute and memory balance.

### 4.3.1 Amdahl's Law of LLMs.

Figure 7 illustrates the Amdahl's Law analysis for the OPT models with 2048 sequence length deployed on the edge TPU setup. The results indicate that, unlike in the cloud setup, across all cases, enhancing the projection layers leads to significantly greater overall improvements in the entire model. Conversely, improving the attention layers yields only marginal gains. This suggests that 1-bit LLM approaches targeting optimization of projection layers are significantly more efficient in the edge setup. Therefore, a promising research direction would be to focus on developing efficient custom hardware for implementing extremely quantized projection layers, rather than concentrating on algorithmic and hardware innovations to enhance computation in the attention heads.

## 5 Conclusion and Future Work

The proposal of 1-bit LLMs has opened several research avenues, including the development of custom hardware for 1-bit LLMs as well as algorithmic innovations to enhance aspects of LLM computation that 1-bit LLMs cannot address. Here, we aimed to provide a roadmap to avoid fundamentally incorrect or inefficient research goals in this field. We introduced the concept of Amdahl's Law of LLMs, which helps determine how partial improvements from 1-bit LLMs translate into overall model enhancements. We conducted extensive evaluations across different LLMs with various hyperparameters to identify relevant patterns. The results reveal a significant dependency of 1-bit LLM efficacy on model sizes, hyperparameters, and hardware configurations. Key findings include: (i) 1-bit LLM paradigms have limited impact on smaller language models, particularly when the context length is large, (ii) for medium-sized LLMs, 1-bit LLM methods show benefits, but further algorithmic innovations are needed to enhance parts that 1-bit LLM approaches cannot improve, and (iii) for large-scale LLMs, extreme quantization from 1-bit LLMs alone can improve the majority of computations, in some cases by more than 99%. While the observation that linear projection layers increasingly dominate compute at larger model scales may be expected from dimensional analysis, our contribution lies in quantifying this shift precisely across a wide range of real-world LLM configurations (OPT, GPT, LLaMA) using cycle-accurate simulations on custom-designed TPU architectures. This approach moves beyond theoretical speculation and establishes an empirical foundation for throughput bottlenecks under extreme quantization regimes.

The proposed research opens up several promising directions for future work, including: (i) expanding the design space to encompass emerging model variants, such as multi-query attention, Mixture-of-Experts (MoE),

and linear attention, and studying their impact on $F$, the overall bottleneck profile, and optimal hardware allocation; and (ii) integrating memory-efficient attention mechanisms (e.g., FlashAttention (Dao et al., 2022), PagedAttention (Kwon et al., 2023)) and refining the analysis within hardware-specific implementations to evaluate their effects on memory access patterns.

## Acknowledgments

This work is supported by the National Science Foundation (NSF) under grant numbers 2340249 and 2409697.

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

# A  TPU Dataflow Analysis

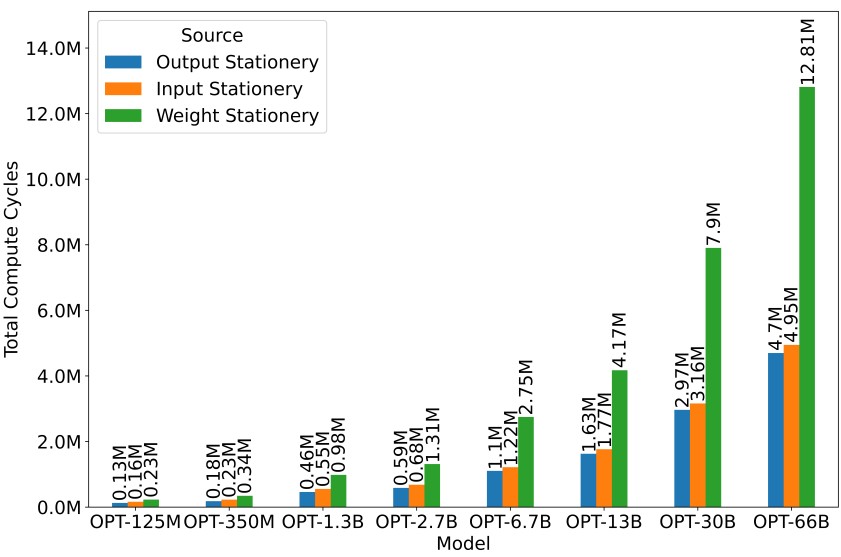

(a): Cloud Setup

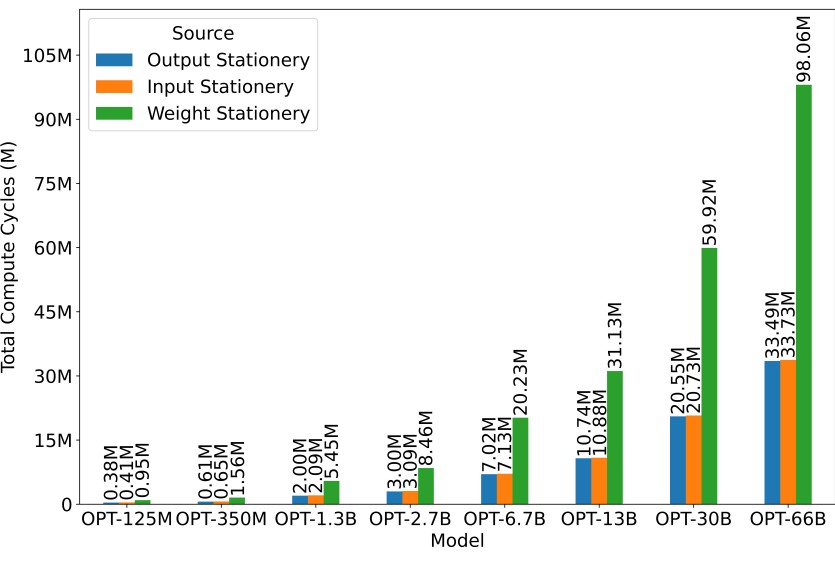

(b): Edge Setup

Figure 8: Total cycles for different dataflow of OPT models.

All experiments were conducted on a computing setup with a single node, 48 cores, and 200 GB of memory. The Scale-Sim V2 tool was installed, and experiments were executed using specific configurations for both cloud and edge setups.

# B GPT and LLaMA Results

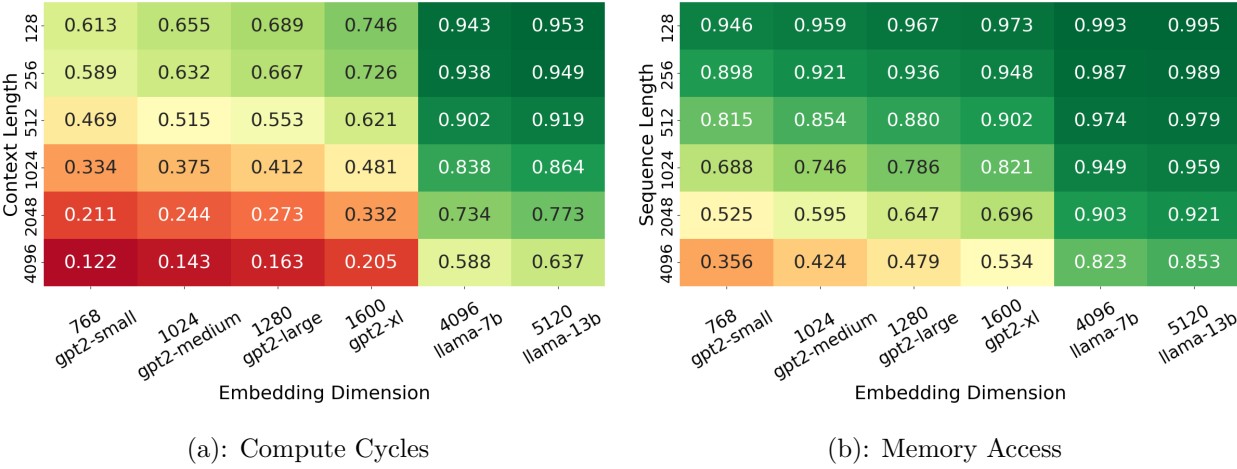

(a): Compute Cycles

(b): Memory Access

Figure 9: Fraction of MatMul-free operations in the GPT and LLaMA models for the cloud setup.

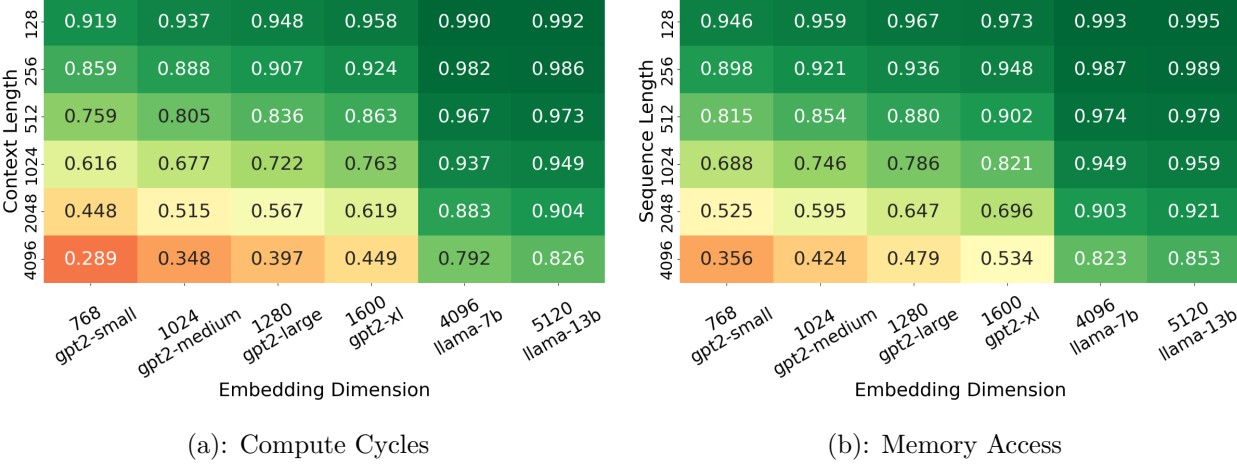

(a): Compute Cycles

(b): Memory Access

Figure 10: Fraction of MatMul-free operations in the GPT and LLaMA models for the edge setup.

All experiments were conducted on a computing setup with a single node, 48 cores, and 200 GB of memory. The Scale-Sim V2 tool was installed, and experiments were executed using specific configurations for both cloud and edge setups.

## C   Amdahl's Law Analysis of OPT Models for Various Sequence Lengths

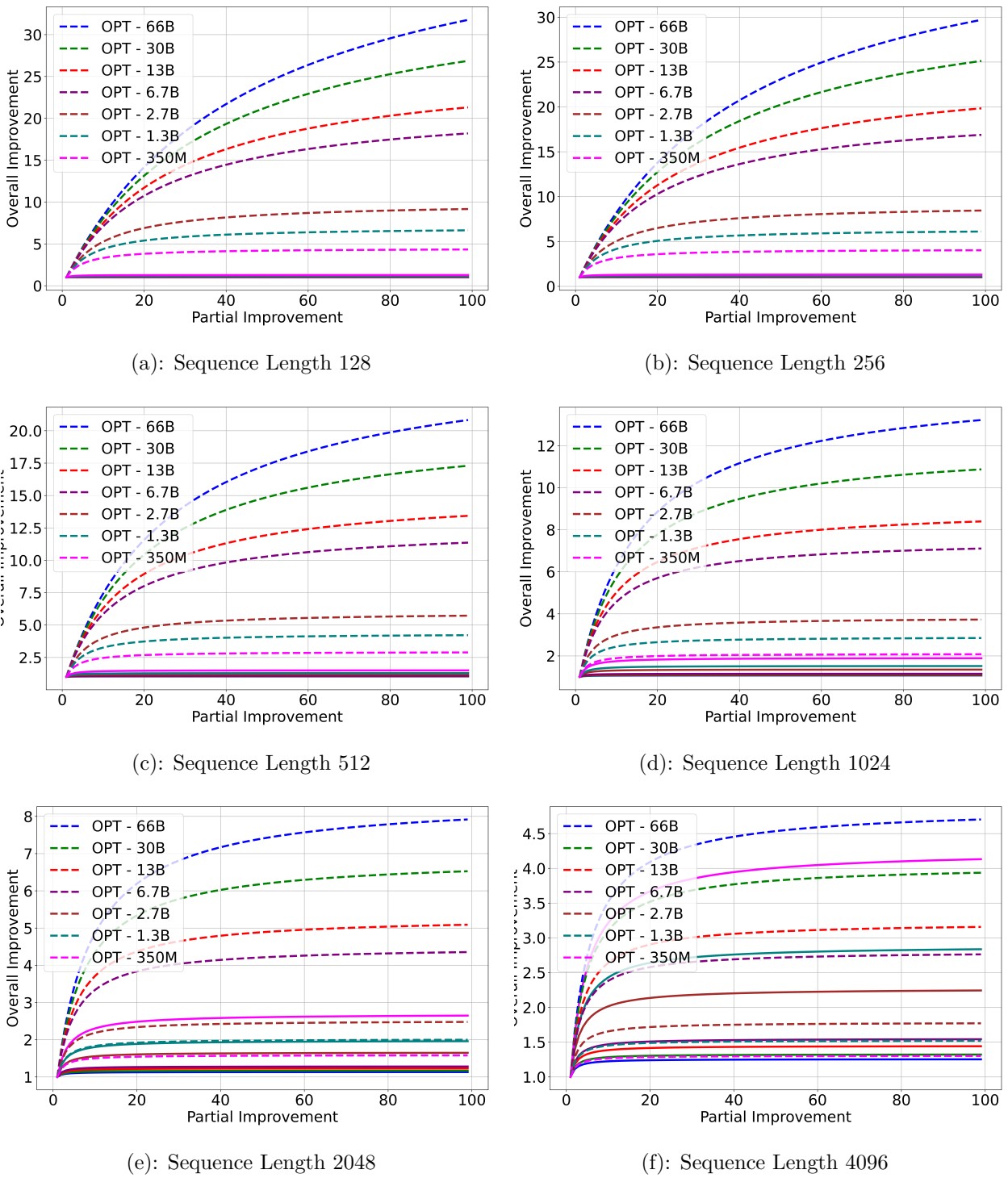

Figure 11: Amdahl's law for different sequence lengths in the OPT models for the cloud setup.

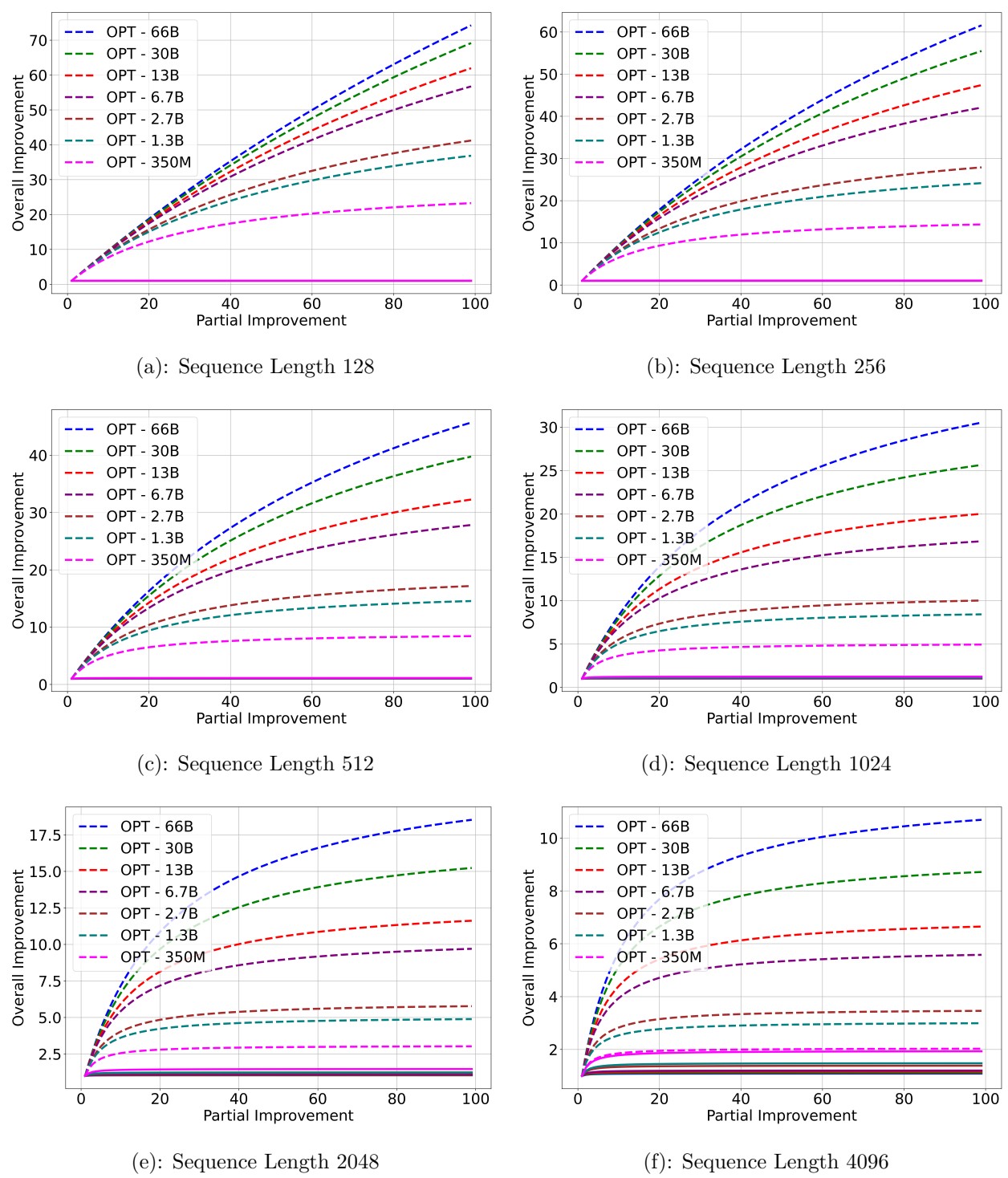

Figure 12: Amdahl's law for different sequence lengths in the OPT models for the edge setup.

All experiments were conducted on a computing setup with a single node, 48 cores, and 200 GB of memory. The Scale-Sim V2 tool was installed, and experiments were executed using specific configurations for both cloud and edge setups.

## D    GPU Results

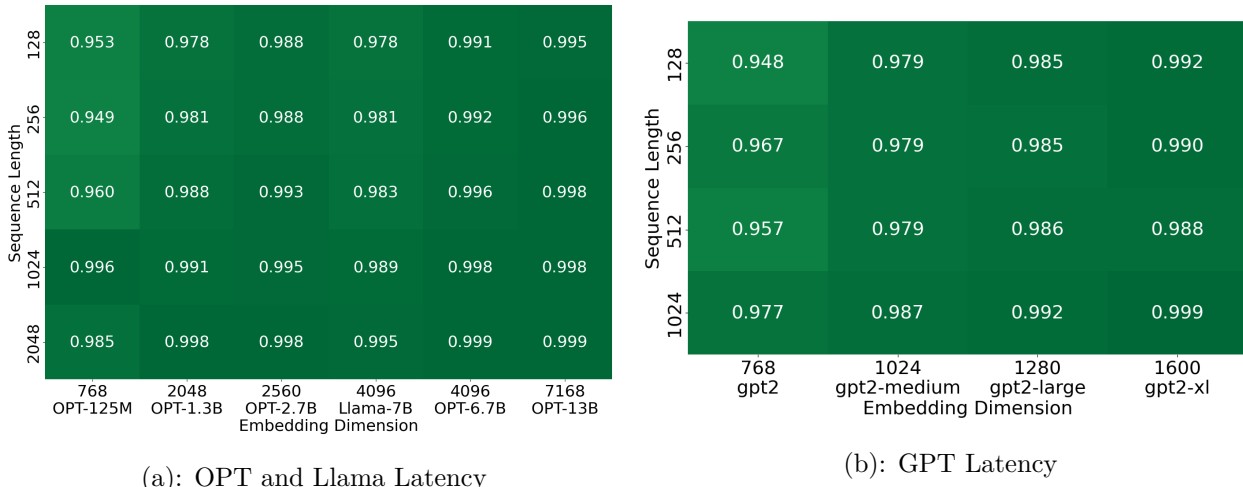

(a): OPT and Llama Latency

(b): GPT Latency

Figure 13: Percentage of latency of projection layers measured using GPU. All experiments were conducted on a computing setup with a single node, 40 cores, NVIDIA Tesla V100 with 32 GB of RAM.

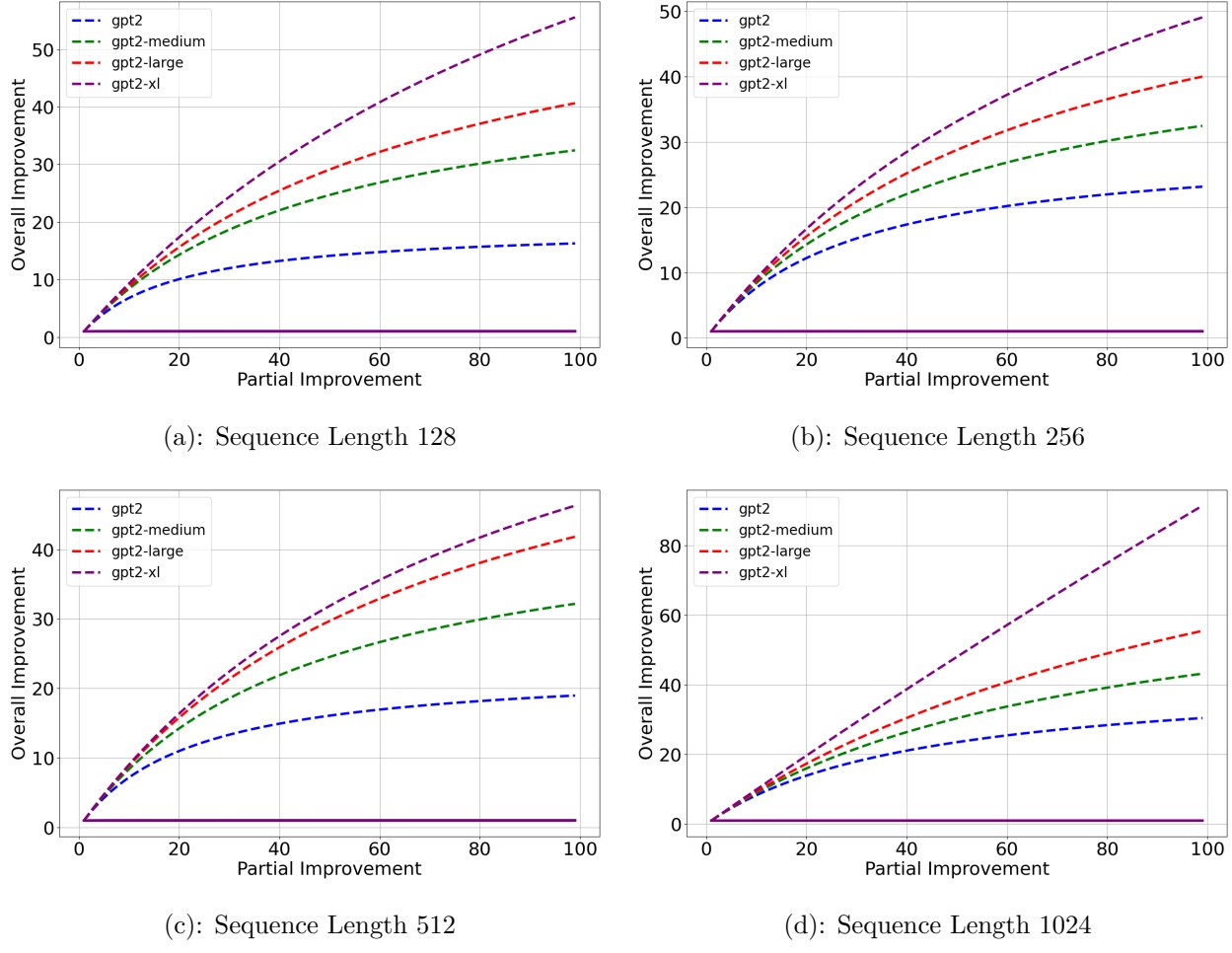

(a): Sequence Length 128

(b): Sequence Length 256

(c): Sequence Length 512

(d): Sequence Length 1024

Figure 14: Amdahl's law analysis on GPU of GPT models.

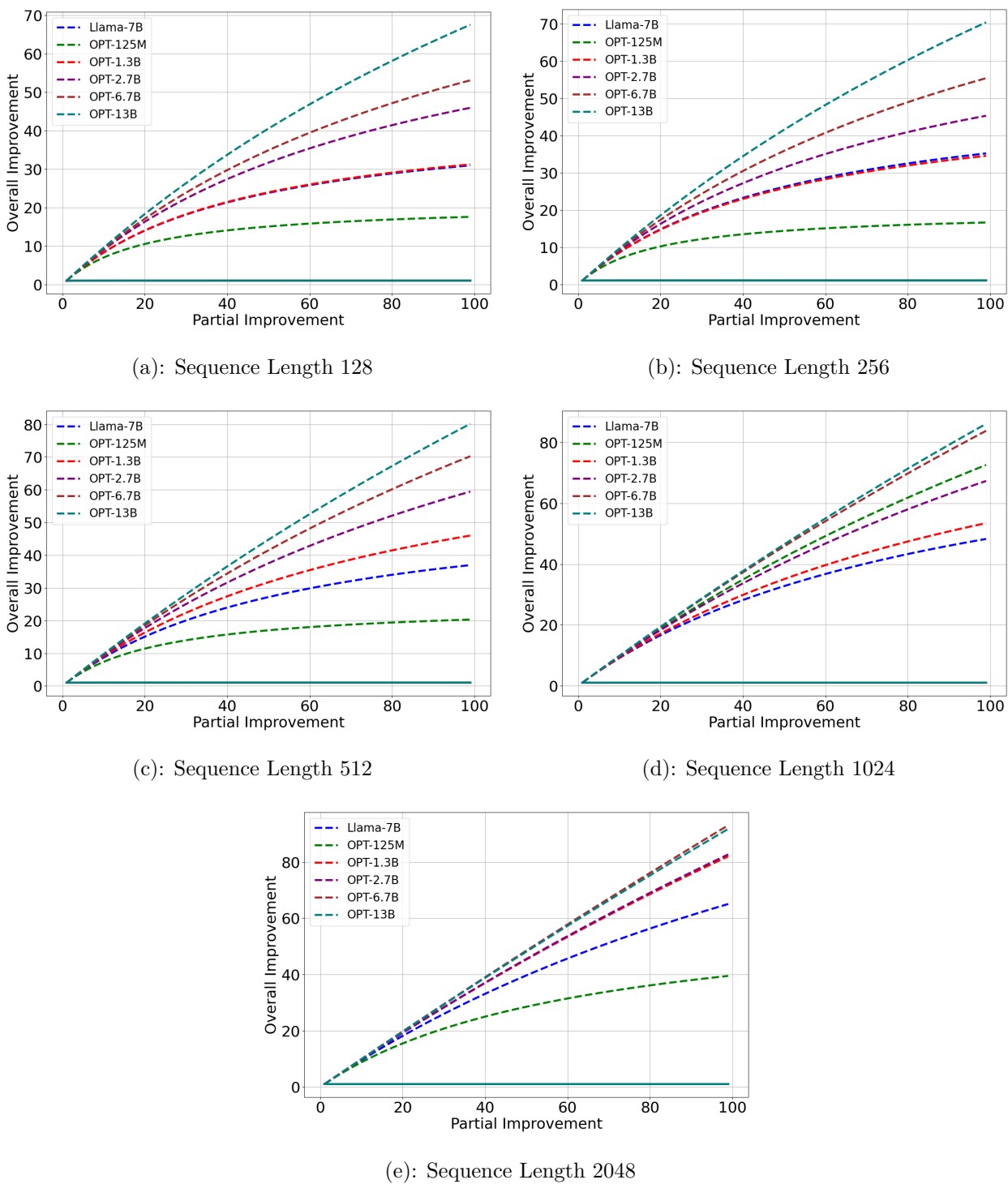

Figure 15: Amdahl's law analysis on GPU of OPT and LLama models.

The GPU latency analysis is intended to contextualize the TPU-centric findings and highlight that even on GPU-optimized transformer kernels the projection layers remain the dominant latency contributor, especially in large models. For instance, Figure 12(a) shows that in LLaMA and OPT models, projection layers account for more than 95% of total latency, reinforcing our claim that quantization of projection layers yields substantial benefits across hardware backends.

# E    Analysis of Attention Head Count

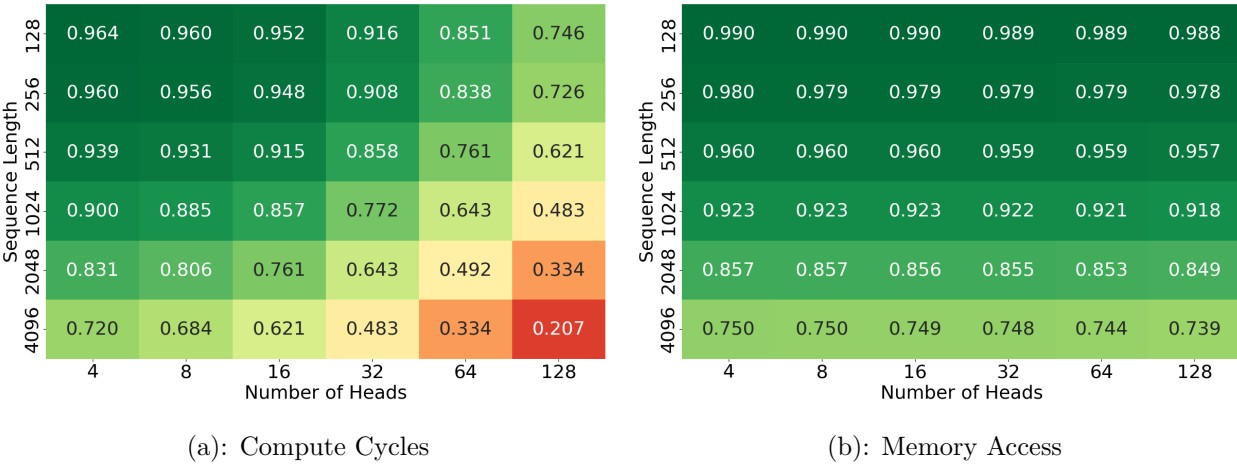

(a): Compute Cycles                         (b): Memory Access

Figure 16: **Cloud TPU Setup.** Heatmaps illustrate the fraction of MatMul-Free operations in the projection layers across different numbers of attention heads and sequence lengths, with the embedding dimension fixed at $d = 4096$. As the number of heads increases (with fixed $d$), the MatMul dimensions within each attention head decrease, leading to under-utilization of large ($256{\times}256$) systolic arrays in the cloud setup. This results in larger compute cycles in the attention heads, for instance, with $h = 128$, 79.93% of compute cycles occur in the attention heads, while only 20.7% are in the projection layers.

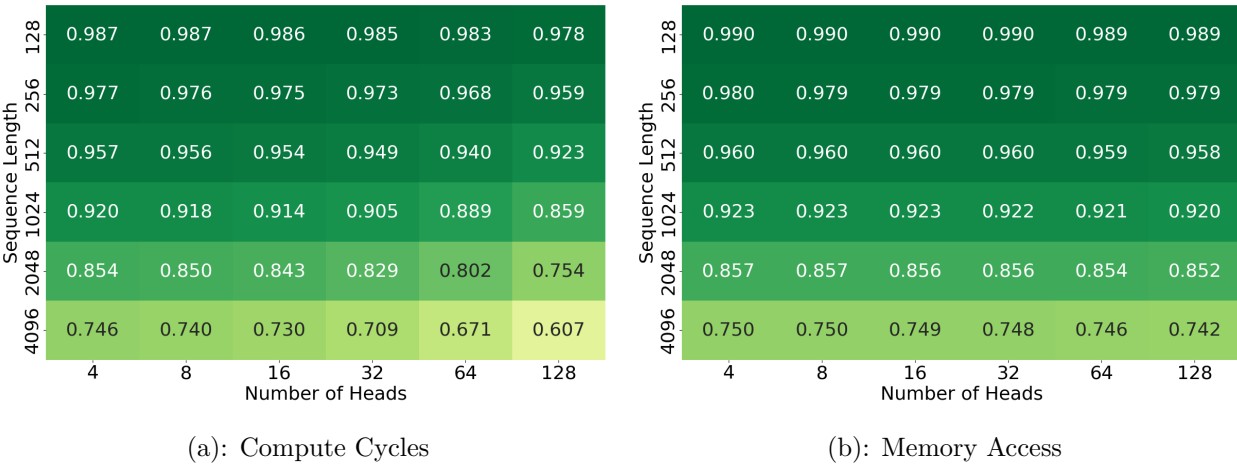

(a): Compute Cycles                         (b): Memory Access

Figure 17: **Edge TPU Setup.** Heatmaps show the ratio of MatMul-Free operations in projection layers across varying attention head counts and sequence lengths, with the embedding dimension fixed at $d = 4096$. On edge TPUs, smaller attention head MatMuls can be efficiently handled by small systolic arrays (e.g., $32{\times}32$) without incurring the overhead of data movement between processing elements seen in larger arrays. As a result, most of the clock cycles still happen in the projection layers.

# F Expanding the Generalization and Applicability of Amdahl's Law for LLMs

While our simulations currently focus on TPUs as a representative hardware baseline, we recognize that modern LLM deployment spans a diverse range of hardware targets. These include general-purpose GPUs (such as NVIDIA's A100 and H100), reconfigurable platforms like FPGAs, application-specific integrated circuits (ASICs), and emerging memory-centric architectures such as processing-in-memory (PIM) accelerators. Each of these platforms introduces unique tradeoffs in compute throughput, memory bandwidth, energy efficiency, and support for sparsity or quantization.

To enable a more general and hardware-agnostic analysis, the proposed Amdahl's Law of LLM can be expanded to explicitly factor in the non-uniform acceleration characteristics of various LLM components. Instead of assuming a monolithic speedup across the entire workload, this formulation distinguishes among different layers or operations within the model, specifically, the projection layers, attention heads, and feedforward networks (FFNs), and allows their respective speedups to be expressed independently. This can be modeled as:

$$\text{Speedup} = \frac{1}{\text{Non-Quantized Portion} + \frac{f(\text{proj})}{S_d} + \frac{f(\text{head})}{S_h} + \frac{f(\text{FF})}{S_{dff}}} \tag{6}$$

Here, $f(\cdot)$ denotes the fractional contribution of each component to the overall computation time in the baseline (unaccelerated) system. The terms $S_d$, $S_h$, and $S_{dff}$ represent the realized speedup of the projection layers, attention heads, and feedforward components, respectively, on the target hardware. The "Non-Quantized Portion" term accounts for operations that remain unaccelerated either due to algorithmic constraints or hardware limitations (e.g., softmax normalization, residual connections, or embedding lookups). These speedup terms are determined by the characteristics of the underlying hardware architecture. For example, in our case, the speedups reflect the performance of systolic arrays, which vary in size and efficiency across cloud and edge platforms. However, our model is flexible enough to incorporate other architectural paradigms. For instance, PIM-based accelerators may be particularly effective at accelerating projection layers, while systolic arrays or GPUs may offer greater advantages for matrix-heavy feedforward operations. This flexibility enables modeling of hybrid systems, where different layers of the LLM are mapped to specialized hardware for optimal efficiency.

By explicitly decoupling the model architecture from the hardware acceleration characteristics, this formulation allows practitioners to reason about performance tradeoffs in a modular way and extend the analysis to heterogeneous or mixed-precision systems.

