# OpenReview forum: "Amdahl’s Law for LLMs: A Throughput-Centric Analysis of Extreme LLM Quantization"
_TMLR — Accepted by TMLR_

### Review · Reviewer_dcc9 · 2025-06-13

**Summary Of Contributions:**

The authors present an analysis of the computational benefits of aggressively quantizing the MLP layers of large-scale Transformer models, using simulations of machine-learning accelerator hardware. The key point is that Amdahl's law limits the benefits that can be had based on the model size, because some portion of the computation/overhead is due to the attention blocks, which are harder and less commonly quantized. This leads to conclusions about when and where quantization can have a benefit in language models, with concrete numbers and data for practitioners to reference.

**Audience:**

Yes

**Claims And Evidence:**

No

**Requested Changes:**

This work provides an analysis of LLMs from the perspective of Amdahl's law and its interaction with quantization, but does analyze how different model designs, hardware, and other factors interact thoroughly enough to be broadly useful. My main concerns are these: Does the use of simulation yield any new insights outside of the already-established theory of how LLM performance should scale? And, do the simulations cover enough configurations/model architectures used in practice, with sufficient discussion about the effects of certain design decisions, for the results to be applicable?

I therefore believe the work could benefit from more thorough investigation of different architecture types (both neural-network and hardware). Covering all commonly-tweaked hyperparameters such as attention head size and discussing the effects of architecture changes more thoroughly would make the results of the authors' approach more useful for real-world deployments. Otherwise I believe many readers might question if the results/numbers presented here apply to their use case. For example: while attention can be a bottleneck, the authors do not investigate what happens when the design/shape of this layer is changed, even though this is arguably just as commonly done as changing sequence length or embedding dimension. There is also an opportunity to further research the specific details of how hardware works with these systems, deriving more general conclusions based on hardware-level design decisions that help readers understand Amdahl's law for LLMs overall rather than just as it pertains to the specific hardware discussed in the paper. For example, more hardware than just TPUs use systolic arrays, so it is worth advertising and justifying that the findings here are more broadly applicable. By understanding *why* certain decisions affect the numbers, readers could build an understanding ofthe behavior of Amdahl's law in LLMs in general, which they could then apply broadly.

**Strengths And Weaknesses:**

Strengths:
- the authors provide an analysis of how basic model architecture choices (embedding dimension, sequence length) affect the tradeoffs
- the authors provide extensive numeric data for the aspects of model/hardware design they explored, based on their simulations
- the authors are careful to pay attention to memory access costs, as these too can be a bottleneck or source of inefficiency in DL systems
- the discussion offers clear, actionable advice for when to use this kind of quantization, and the simulations offer a realistic perspective on what the benefits really are

Weaknesses:
- I am not sure if some of the conclusion here provides useful insights that could not be inferred from Amdahl's law; in other words. It seems like a somewhat trivial conclusion that the speedup from quantization would be limited by those operations that could not be quantized.
- I believe the value here is from the authors' use of simulation, because their numbers appear to be derived from a model of real-world inefficiencies, idiosyncrasies, and processes that otherwise make them hard to guess a-priori. But the paper does not apply much effort towards explaining these intricacies, and what interesting insights - if any - the authors learned from simulations compared to plugging estimates of model FLOPs directly into the formula for Amdahl's law. For example, were there any cases where the hardware struggled with a particular architecture, or where energy/throughput was worse than expected because of how the Transformer architecture/scale interacted with the systolic array behavior? Did certain implementation details affect the numbers? The base idea of Amdahl's law is obvious, but findings about the quirks of hardware and models in simulation might provide more ideas about why these tradeoffs occur and what could be done about them. Currently, it is unclear to me what new knowledge, if any, is gained by doing the simulations.
- Relatedly, because others are aware of the limitations of accelerating non-attention operations in Transformers, people have tried implementing alternative architectures to work around these issues. What would be the results of simulating on multi-query attention, sparse-expert, or linear-attention models? These I think would be useful takeaways from an Amdahl's-law-style analysis: these are much newer architectures claiming to produce an advantage by reducing some sort of attention-related overhead, for which there is far less rigorous work studying their real-world benefits based on benchmarks and their effects on hardware utilization. Some of these are key advances in the design of Transformer models that are now used a lot in practice, and so I believe that investigating them within this framework is necessary for the results of this work to be useful to practitioners.
- Models may reduce the memory intensiveness of multi-head attention by using larger and fewer attention heads. In terms of studying the effect of transformer design, the number of attention heads is an important hyperparameter that I do not believe is sufficiently considered in this work, given that attention itself can often be the main performance bottleneck of interest. Indeed, hyperparameters like attention heads do change in the models the authors simulate, but appear not to have their effects analyzed explicitly, such as in Figure 4/5. Also, when plotting results in terms of embedding size and sequence length, it is hard to disentangle whether the final numbers are affected by the change in these two parameters or change in number of attention heads.

The result of these limitations is that the analysis of specific hardware simulation and only certain architecture variables in LLMs mean that the scope of the conclusions' applicability to real-world cases is limited; readers may have a hard time figuring out if the results here are actually predictive for their particular setup or choice of hyperparameters; there is not enough evidence and cases considered to make a convincing general argument about what Amdahl's law looks like for LLMs in different configurations of model and hardware, in my opinion.

---

> ### Author Response · Authors · 2025-07-25
>
> $\text{\textbf{Concern 1: Amdahl’s Law application appears trivial without deeper insight from simulation}}$
>
> $\text{\textbf{Response}}$: We understand the concern and would like to emphasize that our contribution is not simply restating Amdahl’s Law, but instantiating it with real model and hardware characteristics derived from cycle-accurate simulations of TPU-like accelerators.
>
> The detailed modeling of three systolic array dataflows (IS, WS, OS) enables us to produce fine-grained measurements of compute and memory partitioning between projection and attention operations across varied model configurations. The resulting empirical values of F vary non-linearly with model size, embedding dimensions, and sequence length, insights that are not obvious from theory alone.
>
> Moreover, our simulation findings reveal that attention bottlenecks diminish in high-parameter LLMs, especially under short-to-moderate sequence lengths, a nuance that could not be captured without hardware-level modeling. We will clarify these insights to make the added value of our simulations more evident.
>
> $\text{\textbf{Comment 2: Lack of architectural diversity (e.g., multi-query attention, sparse expert models)}}$
>
> $\text{\textbf{Response}}$:This is an excellent point. While our current focus is on decoder-only transformer architectures following standard designs (OPT, GPT, LLaMA), we agree that emerging variants such as multi-query attention, Mixture-of-Experts (MoE), and linear attention offer promising alternatives with different compute and memory profiles.
> Our current analysis provides a foundational framework  by extending the model-specific parameters in the function:
>
> $F(d, d_{\text{ff}}, h, l) \sim A \cdot d + B \cdot d_{\text{ff}} + C \cdot h + D \cdot l$
>
> We can adapt Amdahl's Law to accommodate such architectures, provided their operational primitives are defined. In the next iteration, we can broaden the design space to include attention variants, examining how these affect F, the overall bottleneck profile, and the optimal hardware allocation.
>
> $\text{\textbf{Comment 3: Insufficient attention to the role of attention heads and hyperparameters}}$
>
> $\text{\textbf{Response}}$: We agree that the role of attention head count and its interaction with model scale and compute bottlenecks deserves more prominence. While we vary head counts implicitly via model hyperparameters (see Table 2), we acknowledge that a more isolated analysis of how changes in head count affect the attention-vs-projection balance would be beneficial.
>
> The number of attention heads directly affects the granularity and shape of the K/Q/V matrices and their associated MatMuls. For instance, increasing h reduces the per-head dimensionality d/h which can change how efficiently these operations map to systolic arrays especially on edge hardware with constrained array size (e.g., 32×32).
> We will add targeted plots and discussion in a future revision to disentangle this hyperparameter's effect and show how it modulates the split between MatMul and MatMul-free regions, and hence the throughput benefits of 1-bit LLMs.
>
> $\text{\textbf{Comment 4: Limited generality and applicability to reader-specific configurations}}$
>
> $\text{\textbf{Response}}$: We provide a roadmap for generalization and will add an appendix to guide future research across diverse hardware platforms. For example, PIM-based accelerators can improve $S_{d}$ and $S_{dff}$
> ​ but not the attention heads, while FlashAttention specifically improves $S_{h}$
> . We also include GPU results to demonstrate broader applicability.
> To support this, we introduce a hybrid speedup model that decouples hardware characteristics from model architecture:
>
> $
> \text{Speedup} = \frac{1}{\text{Non-Quantized portion} + \frac{f(\text{proj})}{S_d} + \frac{f(\text{head})}{S_{h}} + \frac{f(\text{FF})}{S_dff}}
> $
>
> This factorized, nonlinear formulation enables interpretability and adaptability across hardware backends.

---

> ### Author Response · Authors · 2025-07-28
>
> **For comment 1, we added the following text in section 3.3:**
>
> “This formulation enables practitioners to reason about speedup potential (Spartial) as a function of hardware-specific parameters (e.g., systolic array size) and model hyperparameters such as embedding size d, context length l, feedforward dimension dFF, and number of attention heads h.”
>
> **And the text below in section 5 :**
>
> “While the observation that linear projection layers increasingly dominate compute at larger model scales may be expected from dimensional analysis, our contribution lies in quantifying this shift precisely across a wide range of real-world LLM configurations (OPT, GPT,
> LLaMA) using cycle-accurate simulations on custom-designed TPU architectures. This approach moves beyond theoretical speculation and establishes an empirical foundation for throughput bottlenecks under extreme quantization regimes.”
>
>
> **For comment 2, we added the following in Section 5:**
>
> “The proposed research opens up several promising directions for future work, including: (i) expanding the design space to encompass emerging model variants, such as multi-query attention, Mixture-of-Experts (MoE), and linear attention, and studying their impact on F , the overall bottleneck profile, and optimal hardware allocation; and (ii) integrating memory-efficient attention mechanisms (e.g., FlashAttention Dao et al. (2022), PagedAttention Kwon et al. (2023)) and refining the analysis within hardware-specific implementations to evaluate their effects on memory access patterns.”
>
> **For comment 3, we added the below paragraph to Section 4.3 along with a new Appendix E that includes new simulation results and analysis of the impact of attention head MatMul dimensions on the compute and memory balance.**
>
> “Additionally, the cloud setup faces underutilization of processing elements when executing smaller MatMuls in attention heads, whereas the edge setup rarely encounters this issue. This helps explain the observed difference in the compute cycle fraction (F ) between edge and cloud environments, even though the memory access patterns remain largely similar across both. We include additional experiments in Appendix E, where the embedding dimension is fixed (d = 4096) and the number of attention heads is varied, to isolate the impact of attention head MatMul dimensions on the compute and memory balance.”
>
>
> **For comment 4, we added the below text to the last paragraph of Section 3.3, along with a new Appendix F.**
>
> “In Appendix F, we provide a more generalized variation of the proposed Amdahl’s Law of LLM to enable a more general and hardware-agnostic analysis.”

---

### Review · Reviewer_VFaP · 2025-06-26

**Summary Of Contributions:**

This paper presents an analysis on the relative impact of linear layer operations vs. attention matmuls in 1-bit quantized transformers. With binary or ternary weights (1-2 bits), linear layer operations can be replaced by more efficient additions and substraction, and are thus denoted "MatMul-free". The authors apply Amdahl's law to estimate how an improvement in these MatMul-free operations vs. attention matmuls would translate in overall model improvement, thus drawing model-dependent recommendations on where optimization efforts should be placed. Compute cycles and memory accesses are examined across 3 model architectures (GPT, OPT, Llama) of varying sizes, with primary focus on 2 TPU setups. GPU results are provided in appendix.

**Audience:**

No

**Broader Impact Concerns:**

Not discussed in the manuscript. No concerns on my end.

**Claims And Evidence:**

No

**Requested Changes:**

Could the authors please comment on the weaknesses brought up above?

**Strengths And Weaknesses:**

**strengths**
- the topic of improving LLM performance via quantization and related optimization strategies is of interest to the audience of this journal
- the paper is easy to follow, and for the most part well organized

**weaknesses**
- in my opinion, innovation is very limited: the observation that linear layer operations are more dominant for larger model sizes and smaller sequence lengths can be derived from basic mathematical considerations on tensor sizes, and are thus in line with expectation of practitioners. Amdahl's law application does provide a quantitative expectation on the improvement, but it is simply being applied to the problem at hand, with no particular "adaptation" nor being "tailored" (unlike the authors' claims). TPU measurements are of interest but I have concerns regarding the generality of the results (see next point). Consequently, the practical value of this work is also very limited, in my view
- memory access are strongly impacted by the choice of attention implementation (such as various memory-efficient variants on TPU or, on GPU, scaled dot product attention, flash attention, paged attention, etc.). However, this aspect is not discussed
- the implications of using Cloud vs. Edge setup for TPU are not well explained, beyond listing systolic array size and memory capacity in Table 3: why does the Cloud setup show more pronounced trends in terms of compute, while memory access are almost identical (fig 4 vs fig 6)?
- GPU latency results in appendix D show hardly any dependence and are not being discussed, so it's hard to argue that they contextualize the TPU measurements

**other minor comments**
- fig 4 is unintuitively plotted with the sequence length axis decreasing from bottom to top
- fig 5 and 7 (Amdahl's law plots) are not easy to read, with several overlapped lines and a legend that doesn't include the solid lines
- typo in section 4.3: Figure 10 should be Figure 6

---

> ### Author Response · Authors · 2025-07-25
>
> $\text{\textbf{Comment 1:  Limited Innovation and Amdahl's Law "Adaptation"}}$
>
> $\text{\textbf{Response}}$: We appreciate the reviewer’s concerns and would like to clarify the novelty and methodological rigor underlying our analysis.
>
> While the observation that linear projection layers increasingly dominate compute at larger model scales may be expected from dimensional analysis, our contribution lies in quantifying this shift precisely across a wide range of real-world LLM configurations (OPT, GPT, LLaMA) using cycle-accurate simulations on custom-designed TPU architectures. This approach moves beyond theoretical speculation and establishes an empirical foundation for throughput bottlenecks under extreme quantization regimes.
>
> Our selection of TPUs particularly the cloud setup with large 256×256 systolic arrays  serves a methodological purpose. TPUs offer an architectural lower bound on compute cycles for MatMul-heavy operations due to their data reuse efficiency and tightly coupled memory hierarchies. This enables us to bound the best-case performance of LLM inference and establish a rigorous baseline. Comparisons to less specialized architectures (e.g., GPUs or general-purpose accelerators) would only widen the compute gap and reinforce the importance of our findings regarding the dominance of MatMul-free projections at scale.
>
> Our adaptation of Amdahl’s Law is not a direct application but is parameterized specifically for LLM workloads. We define the improvable fraction using actual ratios of MatMul-free operations (via 1-bit projection layers) obtained from simulation data. This formulation enables practitioners to reason about speedup potential as a function of hardware-specific parameters (e.g., systolic array size) and model hyperparameters such as embedding size d, context length l, feedforward dimension dFF, and number of attention heads h.
>
> We believe this quantitative roadmap is essential to preventing misallocation of optimization efforts (e.g., optimizing attention in models where it contributes marginally to latency) an insight not derivable from Amdahl’s Law in isolation, but rather from its data-driven instantiation tailored to the LLM context.
>
>
> $\text{\textbf{Comment 2 : Memory Access and Attention Variants}}$
>
> $\text{\textbf{Response}}$: Thank you for raising this point. In our study, we deliberately isolate the baseline configuration of self-attention to ensure controlled comparisons between MatMul and MatMul-free segments of the model. Specifically, we analyze memory access cycles under three TPU dataflows (input stationary, weight stationary, output stationary) and select the output stationary configuration for detailed analysis due to its superior reuse of partial sums (Appendix A).
>
> Memory-efficient attention variants introduce algorithmic changes (e.g., fused kernels, tiling strategies) that often depend on the GPU memory hierarchy and kernel-level optimizations. Our simulations, in contrast, abstract away such kernel-level differences to emphasize architectural bottlenecks intrinsic to the model, such as matrix dimensions and compute/memory ratios. We acknowledge this as a promising future direction and agree that incorporating such attention variants could refine the analysis in hardware-specific implementations beyond the scope of TPUs.
>
> $\text{\textbf{Comment 3: Cloud vs Edge TPU Behavior}}$
>
>
> $\text{\textbf{Response}}$: This divergence stems from the scaling characteristics of the two setups. In cloud TPUs (256×256 arrays), large matrix multiplications in projection layers are efficiently handled due to ample compute and memory bandwidth, leading to pronounced compute savings for MatMul-free sections. In contrast, edge TPUs (32×32 arrays) suffer degraded efficiency on large projections due to limited parallelism and SRAM, capping compute benefits despite similar memory access trends.
> Additionally, the cloud setup faces underutilization of processing elements when executing smaller MatMuls (e.g., 32×32) in attention heads, whereas the edge setup rarely encounters this issue. This further explains the difference in the fraction (F) between edge and cloud setups beyond the scaling argument.
>
>
> $\text{\textbf{Comment 4: GPU Latency Results Lack Discussion}}$
>
> $\text{\textbf{Response}}$:
> We appreciate the opportunity to clarify. The GPU latency analysis in Appendix D is intended to contextualize the TPU-centric findings and highlight that even on GPU-optimized transformer kernels the projection layers remain the dominant latency contributor, especially in large models. For instance, Figure 13 (a) shows that in LLaMA and OPT models, projection layers account for more than 95% of total latency, reinforcing our claim that quantization of projection layers yields substantial benefits across hardware backends.
>
> We acknowledge that the discussion can be expanded in the main text to clarify this insight and demonstrate the hardware-agnostic value of our Amdahl’s Law adaptation.

---

> > ### Comment · Reviewer_VFaP · 2025-07-26
> > **Response**
> >
> > I thank the authors for their responses and clarifications. Could you please share the revised version of the manuscript, possibly highlighting the changes in different color?

---

> ### Author Response · Authors · 2025-07-28
>
> **For comment 1, we added the following text in section 3.3:**
>
> “...This formulation enables practitioners to reason about speedup potential (Spartial) as a function of hardware-specific parameters (e.g., systolic array size) and model hyperparameters such as embedding size d, context length l, feedforward dimension dFF, and number of attention heads h.”
>
> **And the text below in section 5 :**
>
> “While the observation that linear projection layers increasingly dominate compute at larger model scales may be expected from dimensional analysis, our contribution lies in quantifying this shift precisely across a wide range of real-world LLM configurations (OPT, GPT,
> LLaMA) using cycle-accurate simulations on custom-designed TPU architectures. This approach moves beyond theoretical speculation and establishes an empirical foundation for throughput bottlenecks under extreme quantization regimes.”
>
> **For comment 2, we added the following text in Section 5:**
>
> “The proposed research opens up several promising directions for future work, including: (i) expanding the design space to encompass emerging model variants, such as multi-query attention, Mixture-of-Experts (MoE), and linear attention, and studying their impact on F , the overall bottleneck profile, and optimal hardware allocation; and (ii) integrating memory-efficient attention mechanisms (e.g., FlashAttention Dao et al. (2022), PagedAttention Kwon et al. (2023)) and refining the analysis within hardware-specific implementations to evaluate their effects on memory access patterns.”
>
> **For comment 3, we added the following text in section 4.3:**
>
> “The difference between the cloud and edge setups stems from their scaling characteristics. In cloud TPUs (256×256 arrays), large MatMuls in projection layers are efficiently handled due to ample compute and memory bandwidth, leading to pronounced compute savings for MatMul-free sections. In contrast, edge TPUs (32×32 arrays) suffer degraded efficiency on large projections due to limited parallelism and SRAM, capping compute benefits despite similar memory access trends.
>
> Additionally, the cloud setup faces underutilization of processing elements when executing smaller MatMuls in attention heads, whereas the edge setup rarely encounters this issue. This helps explain the observed difference in the compute cycle fraction (F ) between edge and cloud environments, even though the memory access patterns remain largely similar across both. We include additional experiments in Appendix E, where the embedding dimension is fixed (d = 4096) and the number of attention heads is varied, to isolate the impact of attention head MatMul dimensions on the compute and memory balance.”
>
> **For comment 4, we added the following text in Appendix D :**
>
> “The GPU latency analysis is intended to contextualize the TPU-centric findings and highlight that even on GPU-optimized transformer kernels the projection layers remain the dominant latency contributor, especially in large models. For instance, Figure 12(a) shows that in LLaMA and OPT models, projection layers account for more than 95% of total latency, reinforcing our claim that quantization of projection layers yields substantial benefits across hardware backends.”

---

### Review · Reviewer_mZtb · 2025-07-12

**Summary Of Contributions:**

This paper introduces an adaptation of Amdahl's Law, to provide a quantitative framework for analyzing the throughput limits of 1-bit LLMs. Through experiments on different models and hardware platforms, the authors reveal that the primary performance bottleneck shifts between attention heads and projection layers depending on model size and hardware. This analysis provides a roadmap for focusing future research on the more impactful component.

**Audience:**

Yes

**Claims And Evidence:**

Yes

**Requested Changes:**

1. Clarify the assumed quantization scheme or methods.

2. Include a sensitivity analysis for partial speedup improvement.

**Strengths And Weaknesses:**

Strengths:

1. The paper provides a simple yet general quantitative framework to reason about the performance limits of partial model optimization.

2. The study is supported by extensive experiments covering different LLMs across the OPT, GPT, and LLaMA families, with varying sequence lengths. Analyzing performance on two distinct, custom-designed hardware platforms (cloud and edge) further strengthens the conclusions.

3. The results provide a clear and valuable insights. The key takeaway—that the optimal focus of optimization shifts from attention heads to projection layers depending on model size and hardware—is a practical insight that can help guide research efforts.

Weaknesses:

1. The paper does not explicitly state whether its analysis assumes weight-only or weight-and-activation quantization. While it mentions that activations remain in 8-bit precision in some 1-bit models, a clearer definition of the assumed quantization scheme (e.g., W1A8 or W1A16) is needed.

2. The analysis focuses heavily on calculating the fraction of the model that can be improved (F) but states that analyzing the speedup factor of that part is beyond its scope. The paper would be strengthened by a sensitivity analysis or discussion on how different, realistic speedup values for MatMul-free hardware might alter the conclusions.

---

> ### Author Response · Authors · 2025-07-25
>
> $\text{\textbf{Comment 1: Clarify the assumed quantization scheme}}$
>
> $\text{\textbf{Reviewer Concern: The paper doesn't clearly specify the quantization scheme (e.g., W1A8 or W1A16), despite referencing 1-bit LLMs.}}$
>
> $\text{\textbf{Response:}}$
> Thank you for highlighting this point. We clarify that our work assumes W1A8, W2A8 quantization as the baseline quantization schemes throughout the analysis. Specifically, projection weights are represented using binary or ternary quantization (i.e., 1-2 bits for weights), while activations are maintained at 8-bit integer precision (INT8). This reflects the implementation principles of systems like BitNet and BitNet 1.58, which preserve accuracy by keeping activations in moderate precision while aggressively quantizing weights.
> We will update the manuscript to clearly and explicitly define this assumption early in Section 3, where we describe our system model and quantization methodology.
>
> $\text{\textbf{Comment 2: Sensitivity analysis for partial speedup}}$
>
> $\text{\textbf{Reviewer Concern: A discussion of how different speedup levels for MatMul-free operations would impact conclusions is missing.}}$
>
> $\text{\textbf{Response}}$:
> We appreciate the reviewer’s suggestion and agree that exploring a range of practical Spartial values is essential for interpreting the broader utility of our framework. While our paper focuses on quantifying F, the improvable fraction, we acknowledge that realistic speedup values (Spartial) depend on the hardware backend (e.g., binary hardware accelerators, emerging computer and memory architectures like processing-in-memory (PIM)).
> To that end, our Amdahl's Law plots already span a wide range of Spartial∈[1,100], allowing readers to interpolate performance benefits under different assumptions. However, we agree that a dedicated sensitivity analysis, incorporating empirical or theoretical estimates of Spartial from contemporary hardware, would enhance the paper’s practical relevance. We will include such an analysis in the next iteration, particularly referencing PIM-based designs which have demonstrated 10×–100× speedups on MatMul operations in projection-heavy workloads.

---

> > ### Author Response · Authors · 2025-07-28
> >
> > **For comment 1, we added the following text in section 4.1 :**
> >
> > “Our work assumes W1A8, W2A8 quantization as the baseline quantization schemes throughout the analysis. Specifically, projection weights are represented using binary or ternary quantization (i.e., 1-2 bits for weights), while activations are maintained at 8-bit integer precision (INT8). This reflects the implementation principles of systems like BitNet Wang et al. (2023) and BitNet 1.58 Ma et al. (2024), which preserve accuracy by keeping activations in moderate precision while aggressively quantizing weights.”
> >
> > **For comment 2, we added the following text in section 3.3 :**
> >
> > “…Nonetheless, to provide context, a recent processing-in-memory (PIM)-based hardware design Malekar et al. (2025) targeting the acceleration of projection layers of 1-bit LLMs demonstrated up to an 80× increase in throughput.”

---

> > > ### Comment · Reviewer_mZtb · 2025-08-07
> > >
> > > Thank you for adding the request details. I believe that addressed my concerns.

---

### Decision · Action_Editor_vR6W · 2025-08-28

**Recommendation:** Accept as is

**Additional Comments:**

A recurring point of criticism during the rebuttal was the alleged lack of novelty, as the paper primarily profiles existing quantization strategies. However, TMLR explicitly states that papers are not judged on novelty, but rather on how well their claims are supported. Moreover, while reviewers noted that many of the results may be expected, the paper provides a careful empirical analysis that quantifies these effects.

In summary, I believe the paper should be accepted, as it offers a thorough and well-executed investigation of the considered setting.

**Audience:**

Yes

**Audience Explanation:**

Quantization is an important part of LLM inference and I am sure that the part's of TMLR's audience is interested in this line of research.

**Claims And Evidence:**

Yes

**Claims Explanation:**

Yes the paper provides experimental results across 13 different LLMs on TPUs to support their claims